# Spent Mushroom Substrate Hydrolysis and Utilization as Potential Alternative Feedstock for Anaerobic Co-Digestion

**DOI:** 10.3390/microorganisms11020532

**Published:** 2023-02-20

**Authors:** Gabriel Vasilakis, Evangelos-Markos Rigos, Nikos Giannakis, Panagiota Diamantopoulou, Seraphim Papanikolaou

**Affiliations:** 1Laboratory of Food Microbiology and Biotechnology, Department of Food Science and Human Nutrition, Agricultural University of Athens, 11855 Athens, Greece; 2Laboratory of Food Process Engineering, Department of Food Science and Human Nutrition, Agricultural University of Athens, 11855 Athens, Greece; 3Institute of Technology of Agricultural Products, Hellenic Agricultural Organization “Dimitra”, 1 Sofokli Venizelou Str., 14123 Lykovryssi, Greece

**Keywords:** spent mushroom substrate, chemical hydrolysis, hydrothermal process, anaerobic digestion, biofuels, biomethane, agro-industrial waste valorization, waste-to-energy, circular economy

## Abstract

Valorization of lignocellulosic biomass, such as Spent Mushroom Substrate (SMS), as an alternative substrate for biogas production could meet the increasing demand for energy. In view of this, the present study aimed at the biotechnological valorization of SMS for biogas production. In the first part of the study, two SMS chemical pretreatment processes were investigated and subsequently combined with thermal treatment of the mentioned waste streams. The acidic chemical hydrolysate derived from the hydrothermal treatment, which yielded in the highest concentration of free sugars (≈36 g/100 g dry SMS, hydrolysis yield ≈75% *w*/*w* of holocellulose), was used as a potential feedstock for biomethane production in a laboratory bench-scale improvised digester, and 52 L biogas/kg of volatile solids (VS) containing 65% methane were produced in a 15-day trial of anaerobic digestion. As regards the alkaline hydrolysate, it was like a pulp due to the lignocellulosic matrix disruption, without releasing additional sugars, and the biogas production was delayed for several days. The biogas yield value was 37 L/kg VS, and the methane content was 62%. Based on these results, it can be concluded that SMS can be valorized as an alternative medium employed for anaerobic digestion when pretreated with both chemical and hydrothermal hydrolysis.

## 1. Introduction

The valorization of lignocellulosic biomass by obtaining high-value products, which could be used as building blocks and for new metabolites or bioenergy production, acts as an antidote to the environmental burden and thus is an object of high scientific interest [1,2,3,4,5]. Global production of lignocellulosic biomass, mainly derived from agro-industrial or forestry residues, energy crops and cellulosic wastes, is ~180 billion metric tons annually [4]. The reduction of organic load through cultivation of edible fungi is gaining particular attention [5,6,7,8,9,10]. The cultivation of edible and medicinal mushrooms, especially those belonging to the phylum Basidiomycota, has experienced an increasing growth rate in recent years [8,11,12,13,14]. Commercial mushrooms such as shiitake (*Lentinula* sp.), button mushrooms (*Agaricus* sp.), oyster mushrooms (*Pleurotus* sp.) and wood ear mushrooms (*Auricularia* sp.) are rich in protein, carbohydrates, fiber, vitamins and minerals, while low in fat content [8,14,15,16]. Moreover, bioactive secondary metabolites of mushrooms such as phenolic compounds, β-glucans, sterols, etc., possess health-promoting effects to cure inflammation, hypertension, cancer, hyperlipidemia, hypercholesterolemia and other diseases [14,17,18,19], rendering therefore the cultivation of mushrooms in both liquid and solid-state fermentation configurations one of the pillars of modern industrial, green and red biotechnology [5,17,20].

According to data of the Food and Agriculture Organization of the United Nations (FAOSTAT), global production of mushrooms and truffles in 2020 was approximately 43 million tons and has been steadily increasing especially since 1999. China was the leading producing country with over 40 million tons of production, while the production in Europe was ~1.3 million tons [21]. In Greece, barely 0.97 thousand tons of mushrooms were harvested in 2020 [21], and those belonged mainly to the genera *Pleurotus* sp. and *Agaricus* sp. [22]. Considering that for 1 kg of oyster mushrooms about 4–5 kg of residues, called Spent Mushroom Substrate—SMS, are discarded [23,24,25], one realizes the huge amount of biowaste generated by the mushroom agroindustry. The SMS as a composted organic medium consists of fungal mycelial residues and renewable agricultural residues such as wheat or rice straw, sawdust, sugarcane bagasse, etc., supplemented with manure and gypsum, and is therefore rich in cellulose, hemicellulose and lignin [26,27].

The SMS has been reused for various purposes, such as bioremediation of air, water, and soil contaminants, to manage plant diseases and reduce the effect of pesticides, as mulch, for growing crops (greenhouse and nursery crops, even as a substrate for mushroom reclamation), as feed for livestock, for enzymes’ recovery and also as an alternative renewable carbon source for fermentation and bioenergy production [4,23,26,27,28,29,30,31,32,33,34,35]. In most cases the use of SMS requires physical (grinding, hot water, microwave, wet oxidation, ammonia fiber expansion, supercritical CO_2_ explosion, steam explosion, hydrothermal process), chemical (acid or alkaline hydrolysis, ozonolysis, ionic liquid or organic solvent pretreatment) or biological (microbial or enzymatic) treatment, or a combination of these processes [4,36,37,38,39,40,41,42,43,44,45,46,47,48].

The ever-increasing demand for energy is a major issue in modern society, and research into alternative sustainable forms of bioenergy production is necessary due to the depletion of fossil fuels, as well as the worsening climate change caused by greenhouse gas emissions [49,50]. The “second generation” biofuels, obtained by using lignocellulosic feedstocks (such as the SMS) or generally organic-rich industrial, agricultural and municipal wastewaters are a potential alternative for clean energy production such as biogas (biohydrogen, biomethane), bioethanol, lipids as a platform for biodiesel production, etc., compared to conventional fuels [35,51,52,53,54,55,56,57,58,59,60,61,62,63,64]. Anaerobic digestion (AD) of lignocellulosic biomass to produce biomethane involves a chain of chemical reactions and the presence of a microbial consortium. Practically, polymeric substances are biodegraded through a sequence of metabolic steps consisting of hydrolysis, fermentation (acidogenesis), acetogenesis and methanogenesis [9,25,61,65,66]. The anaerobic digestion process (viz. the biodegradation of organic matter by microbial consortia under the absence of oxygen in order for the synthesis of methane to be performed), being comprised of the four aforementioned stages, is influenced by a plethora of environmental and technological parameters such as the initial C/N ratio of the medium, the pH, the incubation temperature, the organic loading rate, the hydraulic loading rate (for the case of continuous cultures), the presence of recalcitrant inhibitors (i.e., heavy metals, phenolic compounds, etc.), ammonia and/or sulfide into the medium, the physiological state of the inoculum and the potential “optimization” of the inoculum through various techniques (i.e., bioaugmentation, acclimation), etc. [67,68,69,70]. The choice and the utilization of the appropriate substrate represents a very important factor affecting the AD process. Biogas production has been studied for both mono-digestion and co-digestion of SMS, but scientific research mainly focuses on co-digestion of SMS with manure and/or other organic wastes to optimize production [25,58,71,72,73,74,75,76].

The AD process has been used for decades as a way to convert animal manure into valuable chemicals such as methane, hydrogen sulfide, and carbon dioxide. Liao et al. [77] and Wen et al. [78] developed improved hydrolysis methods for enhanced biomethane production from animal manures using dilute acid treatments and other techniques, respectively, while Yang et al. [79] utilized different chemical pretreatments on dairy manure to investigate their effects on biogas production potentials. These studies have demonstrated that animal manure is a promising substrate for biogas production and has potential applications in various ways. The produced biogas is mainly used as a natural fuel for electricity or thermal energy production, injected into the natural gas grid, etc., while the digestate is used as fertilizer [9,80,81,82,83].

The aim of the current study was the optimization of SMS hydrolysis and anaerobic digestion of the hydrolysates for biomethane production. The SMS was pretreated with various types of techniques, and the hydrolysate of the most efficient combination of treatments was used as a feedstock for anaerobic digestion. The SMS hydrolysate was co-digested anaerobically with cattle manure to enhance biomethanation in lab-scale digesters, and the results were discussed.

## 2. Materials and Methods

### 2.1. Determination of SMS Composition

The SMS was kindly provided by a Greek mushroom industry (Green Zin S.A., Sourpi, Magnesia, Greece) after three flushes of *Pleurotus ostreatus* mushroom cultivated through commercial cultivation method [84]. As for the pretreatment, the SMS was dried to constant weight as described below, then ground using a high-speed grinder; the small particles (<1 mm) were sieved and collected for further treatment.

The total solids content (TS) was determined by the oven-drying method. In brief, the SMS was dried at 105 °C for 24 h to constant weight, and the TS content (in g/100 g wet SMS—WS) was calculated based on the Equation (A1) (See Appendix A). Conversely, moisture content (in g/100 g WS) corresponds to the amount of water and residual components volatilized at the mentioned temperature [85]. Ash (g/100 g Dry SMS; DS), the so-called inorganic residue remaining after incineration at 575 °C for 6 h was calculated according to Equation (A2) (See Appendix A) [86]. Therefore, the volatile solids (VS, g/100 g DS) were determined according to Equation (A3) (Appendix A). Total Kjeldahl Nitrogen (TKN, g/100 g DS) was determined through the Kjeldahl method [87] in a KjeltekTM 8100 Distillation Unit (Foss A/S, Hillerød, Denmark) and protein content was then calculated according to Equation (A4) (Appendix A); the multiplication factor results from the assumption that proteins contain approximately 16% nitrogen in their molecule.

To determine the concentration of free sugars, the dried and ground SMS was suspended in water for 3 h. Then the concentration of total reducing sugars was determined and expressed as glucose equivalents (g/100 g DS) through the 3,5-dinitrosalicylic acid (DNS) assay as described by Sumner [88]. The lipophilic substances extraction procedure was performed as described by Sluiter et al. [89]. Briefly, a weighed sample of the dried and ground SMS was placed in a cotton thimble, 190 mL of hexane (as the appropriate solvent for the current process) were added to a pre-weighed spherical flask, and both were placed in the Soxhlet apparatus to start the extraction process. The temperature of the heating mantle was adjusted to achieve 4–5 siphons per hour and the extraction was carried out for 24 h. Subsequently, the extract was evaporated (Flash Evaporator/Rotavapor R-114, BÜCHI Labortechnik AG, St. Gallen, Switzerland) and the residue was weighed. The content of lipophilic substances (g/100 g DS) was calculated according to Equation (A5) (Appendix A).

The determinations of structural carbohydrates, as well as total lignin content, were performed according to Sluiter et al. [90]. The method requires an extract-free sample; therefore, two Soxhlet extractions were performed using water and ethanol as solvents prior to the determination. The dried samples were then subjected to a ‘two-stage’ acid hydrolysis. Specifically, the samples were placed in anaerobic flasks, into which 3 mL of H_2_S0_4_ 72% (*v*/*v*) were added and stirred on a magnetic stirrer at 30 °C for 1 h. Then, the H_2_SO_4_ solution was diluted to a final concentration of 4% (*v*/*v*) by adding high-purity H_2_O to each sample. In the second stage of hydrolysis, the sealed anaerobic vials were placed in the autoclave for 60 min at 121 °C. Along with the samples, the entire process was also performed for a solution with known sugars concentration (Calibration Verification Standard-CVS) to calculate the percentage of sugar loss during acid hydrolysis (Sugar Recovery Standards-SRS). After completion of the process, the samples were vacuum-filtered using pre-weighed glass fiber filters. The filters were placed in pre-weighed ceramic capsules along with the total amount of solid samples and dried to constant weight (80 °C, 24 h). Finally, the samples were placed in the oven at 560 °C for 4 h. The acid-insoluble lignin content (AIL, g/100 g DS) was calculated as the difference of the two weights resulting from drying (80 °C) and incineration (560 °C). The liquid fraction isolated by vacuum filtration was volumetrically measured and the acid soluble lignin’s content (ASL, g/100 g DS) was determined photometrically by UV-Vis spectroscopy (Jasco V-530) at 320 nm. Total lignin (g/100 g DS) was calculated as the sum of the two subfractions.

High-performance liquid chromatography (HPLC) was used to determine quantitatively and qualitatively the structural carbohydrates of the sample, using a Shodex SP0810 column at 60 °C and a flow rate of 0.6 mL/min, with ultrapure water as mobile phase. The pH of each sample was adjusted to a range of 5.0–5.5 by adding CaCO_3_. The samples were centrifuged (9000 rpm, 10 min, 4 °C), the supernatant was filtered (0.02 µm) and the sugars were analyzed by HPLC. The calculation for the correct values of sugar concentrations requires the determination of the sugar recovery rate (SRR) as the quotient of the standard sugar solution concentration (SSSC) before and after autoclave treatment determined by HPLC (see Equation (A6); Appendix A). SRR was used to correct the observed concentration of each sugar in the sample as Sugars Correct Conc. (g/L) (Equation (A7); Appendix A).

To calculate the initial concentration (before hydrolysis) of the carbohydrate polymers (cellulose and hemicellulose), a correction factor was required to include the loss of a water molecule due to the formation of the glycosidic bond. According to the molecular weights (MW), this coefficient was equal to 0.88 (Equation (A8); Appendix A) for pentoses (xylose, arabinose) and 0.90 (Equation (A9); Appendix A) for hexoses (glucose, galactose, mannose). Based on these correction coefficients, the concentration of each polymer (expressed in g/100 g DS) was determined as the result of the corresponding monomer concentration multiplied by the correction factor. The content of cellulose (in g/100 g DS), expressed as glucan equivalents, was determined based on the glucose content, while the sum of the other saccharides (in g/100 g DS) expressed in this case as xylan equivalents, since xylose was the only pentose detected that corresponded to the content of hemicellulose. The composition was determined in two independent SMS samples and each sample was analyzed in duplicate.

### 2.2. Hydrolytic Processes of SMS

The suspensions for the hydrolyses contained 7.5% SMS (*w*/*v*) in dH_2_O (75 g/L), specifically 15 g SMS in 200 mL total volume (in borosilicate glass bottles), hereafter referred to as SMS suspension (SMS-S). The various hydrolysis procedures included an initial pretreatment with or without stirring and/or heating (85 °C or 100 °C), and the most effective procedure was then additionally combined with chemical treatment (H_2_SO_4_ 2% *v*/*v* for acidic or NaOH 2% *w*/*v* for alkaline hydrolysis). The cases are summarized and described in Table 1. The processes lasted for 3 h and samples were taken every hour to monitor the hydrolysis process by determining the concentration of free reducing sugars. The concentration of reducing sugars (g/L) in the hydrolysate was quantitatively determined by the DNS assay. The hydrolysis yield coefficient (HY, %) was calculated as the result of the reducing sugars released (total reducing sugars—free sugars, in g/100 g DS) divided by the sum of the contents of cellulose and hemicellulose (i.e., holocellulose) (as far as only these two polymers consist of sugars) according to the Equation (A10) (Appendix A). The assays were carried out in triplicate.

### 2.3. Optimization of Chemical Hydrolysis through Hydrothermal Process

Hydrothermal treatment (HT) refers to a thermochemical process which results in the decomposition of carbonaceous materials such as lignocellulosic biomass using water under high pressure and temperature conditions [91,92]. The process in this study involves heat treatment of the sample for 1 h at high temperature (140 °C or 150 °C) and high pressure (316.3 or 475.8 kPa, respectively) in autoclave. The current hydrolytic process involved the application of intense thermal treatment, alone or in combination with chemical treatment, to achieve greater polymer disintegration and subsequent increase in free sugar concentration. The effect of various concentrations of chemical reagents was also evaluated. The cases of hydrothermal treatment are summarized and described in Table 2. The concentration of released reducing sugars was assayed through the DNS analysis and the coefficient of hydrolysis yield (%) was calculated. The assays were carried out in triplicate.

### 2.4. Anaerobic Digestion of SMS Hydrolysate

The acidic hydrolysate resulting from the most efficient combination of hydrolytic treatment (0.4% H_2_SO_4_ + heating at 150 °C) and the alkaline one (2% NaOH + heating at 150 °C) were studied as potential substrates for AD in bench-scale improvised digesters (borosilicate glass bottles). The working volume was set at 90% of the total volume since the inoculum occupied 50% *v*/*v* (i.e., 450 mL) and the rest was filled with the hydrolysate. The VS content of the mixture was set at 2.5% and the pH at 7.6 value. Furthermore, the VS originating from the substrate was set at twice the inoculum’s VS (VS_substrate_:VS_inoculum_ 2:1). The batch anaerobic digesters remained in chambers at a constant temperature of 37 ± 1 °C (mesophilic conditions) for 15 days.

In the case of acid hydrolysis, the suspension containing both the solid SMS residues of hydrolysis and the reducing sugars (concentration ≈ 26.9 g/L, derived from hydrolysis of 18.75 g dry, ground SMS in 250 mL dH2O, TS = 7.5%, VS = 6%) was appropriately diluted (from 6% initial VS concentration to 3.33% final concentration, by addition of 200 mL dH_2_O) and was used as digestion feedstock after neutralizing the pH with sodium hydroxide. As for the alkaline hydrolysate, the pulp-like suspension (containing 18.75 g dry, milled SMS in 250 mL dH_2_O, TS = 7.5%, VS = 6%) was also diluted in a similar way as the previous one and was used as digestion feedstock after neutralizing the pH value with hydrochloric acid. Slurry, kindly provided by a local cow dung biogas plant, was used as inoculum. The properties of the inoculum were pH = 8.0, TS = 3.15% *w*/*v*, and VS = 2.1% *w*/*v*; therefore, it was diluted (from an initial concentration of 2.1% VS to a final concentration of 1.67%), so that the content of VS met the experimental specifications, as previously defined.

The biogas digester was equipped with three outlets; one for digestate sampling, one for gas sampling and one for collection and quantification of total biogas, as depicted in Figure 1. Each bottle was flushed with nitrogen gas (2–3 min) to achieve anaerobic conditions. The digestate and biogas samplings were carried out daily. The digestate was analyzed for volatile solids content and pH (pH/mV meter HΙ 8014-Hanna Instruments). The biogas samples containing CH_4_, CO_2_ and other trace gases, such as H_2_, H_2_S, N_2_, CO [trace content 1–5% [93]; approximately 3% is assumed in the equation below] were analyzed using the slightly modified potassium hydroxide assay [94] to indirectly calculate the methane content (MC, %). Briefly, a specific volume of biogas (Volume A) was injected into an airtight borosilicate glass bottle containing saturated KOH solution through a syringe inserted into the inlet tube of the bottle’s cap. The CO_2_ contained in the biogas reacted with the potassium hydroxide to form soluble potassium carbonate (K_2_CO_3_), while the remaining gas consisting mainly of methane was directed into the outlet tube, in which a glass syringe was fitted. The displacement of the piston, which is due to the pressure of the exiting gas, indicates the remaining biogas volume (Volume B). Therefore, the CO_2_ and CH_4_ contents are calculated according to Equations (A11) and (A12) (Appendix A), respectively [94].

The released gas was collected in a gasholder (borosilicate glass gasometer) containing acidified saturated saline solution as the most suitable barrier solution to limit CO_2_ dissolution and total biogas loss according to Walker et al. [95]. Biogas quantification (in mL) was based on the liquid displacement method [96]; in specific, the pressure in the gasholder headspace was gradually increasing, due to the progressive digestion and biogas accumulation. As a result, the gas displaced the acidified liquid from the sealed bottle into an open communicating volumetric measuring cylinder through a tube immersed in the liquid. The volume of the liquid displaced was measured once per day and was considered equal to the total volume of released biogas. According to that, the cumulative biogas production (CBP, expressed in mL) and the biogas productivity (BP, expressed in mL/d) were determined. Cumulative biogas yield (CBY, in L/kg VS) was determined as the cumulative amount of biogas produced per kilogram of volatile solids consumed. The digestions were carried out in duplicate, and each sample was analyzed in duplicate.

## 3. Results—Discussion

### 3.1. SMS Composition

Total solids of the SMS as supplied by the mushroom production plant were 54.7 g per 100 g wet SMS. Volatile solids were 80.15 g per 100 g of dry SMS, and the contents of all constituents are performed in Table 3. The contents of cellulose and hemi-cellulose, which are of particular interest, because hydrolysis of these polymers releases monosaccharides that are assimilated by microorganisms during AD, were 32.18 and 10.45 g/100 g DS, respectively. Comparing the data of SMS and holocellulose contents in the literature with those obtained in this study, we find that the value of cellulose is almost the same, while the hemicellulose content of SMS is slightly lower [24,28,39,66,97].

### 3.2. SMS Hydrolysates

For the hydrolysis of the structural biopolymer carbohydrates (cellulose, hemicellulose) and the release of the monosaccharides, the application of acid or alkaline chemical treatment combined with thermal process was investigated. The results are summarized in Table 4 and the progression of the increase in sugar due to the hydrolytic processes is performed in Figure 2. As far as the samples subjected to alkaline treatment (combined or not with thermal treatment) are concerned, the rheological properties of the suspension changed, as it became more viscous, like a pulp. This result is consistent with the literature, as mild alkaline treatment selectively removes lignin by degrading of ester and glycosidic side chains, without cellulose degradation, whilst increasing porosity and surface area, thereby improving accessibility to microorganisms [45,98]. Alkali-mediated removal of acetyl and uronic acids leads to total hemicellulose depletion, as described by Loow et al. [99] and confirmed by the results of Shetty et al. [42]. On the other hand, alkaline hydrolysis with sodium hydroxide did not increase the concentration of soluble reducing sugars, as also observed by Wu et al. [39] with either sodium or calcium hydroxide solutions. The increase in released sugars, when treated with alkaline chemical agents, could only be achieved by subsequent enzymatic hydrolysis (enzymes such as cellulases and xylanases) [40].

According to the results, the combination of thermal (100 °C) and acidic (H_2_SO_4_, 2%) chemical treatment resulted in the most efficient hydrolysis of SMS (HY = 43.6 %) and the highest concentration of total reducing sugars (22.67 g/100 g DS) in a 3-h long process. On the contrary, no additional free sugars were released, when SMS were treated by heating only, while a decrease was observed in acid hydrolysis without thermal process, which could be attributed to a possible conversion of free sugars to furfurals and hydroxy-methyl furfurals (HMF), the derivatives of pentose and hexose dehydration, respectively [92,100]. According to Wu et al. [39] the combination of acidic treatment and thermal process was the most efficient hydrolytic process, especially when sulfuric acid (2%) was used and temperature was 121 °C; 31.5 g of reducing sugars per 100 g raw SMS were determined, indicating that a higher temperature could increase the hydrolysis efficiency, which is also confirmed by the results of Qiao et al. [28].

The increase in the yield of SMS chemical hydrolysis derived from increasing the temperature (from 85 to 100 °C) led to further investigation of hydrolysis optimization by combining chemical treatment and hydrothermal process at even higher temperatures, especially 140 or 150 °C. The concentration of total reducing sugars resulting from acid chemical hydrothermal treatment for both temperatures tested are presented in Figure 3a. The hydrothermal treatment at higher temperature and pressure increased the concentration of released reducing sugars, compared to the values derived from acid chemical thermal treatment at 100 °C. The highest concentration = 35.8 ± 1.9 g/100 g DS was obtained by the combination of 0.4% H_2_SO_4_ and heating at 150 °C, while the hydrolysis yield of cellulose and hemicellulose was maximum = 74.5 ± 2.1%, according to Figure 3b. The intensive hydrothermal treatment (at 140 or 150 °C) combined with the activity of acidic chemical reagent at concentrations higher than 0.5% resulted in a reduction of the determined released reducing sugars, due to the formation of undesirable furfurals and related compounds, that additionally have an inhibitory effect on the growth of microorganisms during cultivation [92,100,101,102]. The hydrothermal process at 140 or 150 °C combined with alkaline treatment (NaOH 1 or 2%, *w*/*v*) did not release any reducing sugars, but only affected the rheological properties of the sample, as described above.

### 3.3. Biogas Production

The hydrolysate derived from the acid chemical treatment with 0.4% H_2_SO_4_ combined with hydrothermal process at 150 °C, a combination that resulted in the highest hydrolysis yield, was used as AD feedstock. The feedstock containing both the sugars and the solid residues of SMS hydrolysis was mixed with the inoculum (1:1 ratio) after both suspensions were appropriately diluted, as described previously (see: Materials and Methods—Anaerobic digestion of SMS hydrolysate). The graphical demonstration for the 15-day AD of the acidic hydrolysate is presented in Figure 4a,b. Biogas release started after 2 days of microorganisms’ adaptation to the new anaerobic culture conditions. The pH value remained in the range of 6.5–8.0, which is an ideal range for AD. The daily productivity stabilized (33.0 ± 1.7 mL/d) after 10 days. The volatile solids content in the mixture decreased by 1.40 ± 0.08 units to a final value of 1.1%. The conversion of volatile solids to biogas (CBY) peaked at 52.0 ± 2.1 L/kg VS at the end of AD. The total biogas production of this 15-day AD was 653 ± 11 mL, containing 65.0 ± 1.4% biomethane.

The alkaline hydrolysate derived from 2% NaOH treatment and hydrothermal process at 150 °C was diluted as well and mixed with the inoculum (1:1 ratio). The graphical demonstration for the 15-day AD of the alkaline hydrolysate is presented in Figure 5a,b. In this case, biogas production started 5 days after inoculation. The pH value remained also in the range of 6.5–8.0 during AD. The daily productivity stabilized (29.0 ± 1.8 mL/d) after 12 days and the total biogas production of the 15-day AD was 338 ± 8.6 mL. The volatile solids content in the mixture decreased by 0.9 units to a final value of 1.63 ± 0.11% resulting in a biogas yield of 37.0 ± 2.9 L/kg VS and methane content of 62.1 ± 2.6% at the end of AD.

The utilization of the ideal substrate or blend of substrates represents a very important factor affecting the AD process. The valorization of pretreated SMS blended with manure showed interesting results demonstrating the potential of the implicated agro-industrial residues upon the efficiency of this process and indicating the potential of the mixing of feedstocks, that can positively affect the efficacy of the digestion process [67,103,104]. In most cases, not every single type of feedstock meets all the specifications required for AD processes. Anaerobic co-digestion of spent mushroom substrate with various livestock manure increases biogas production in contrast to AD of SMS or manure as mono-substrate [71,72,74,75,76], while co-digestion of cattle manure and SMS supplemented with other feedstocks such as corn stover or sugar mill effluent has also been studied [24,73]. In our study, co-digestion of chemically and thermally pretreated SMS with dairy manure (DM) was investigated (VS ratio SMS:DM = 2:1) and the results were slightly higher than those presented by Gao et al. [76] (CBY ≈ 29 L/kg VS, MC ≈ 35%), when they used the same ratio of SMS to DM, which is probably due to the fact that there was no prior hydrolysis of SMS in their experiments. In the current study, monosaccharides (glucose and xylose) derived from acid chemical hydrolysis of holocellulose were assimilated by the microorganisms during AD and biogas production was not delayed, as delayed when alkaline hydrolysate was digested. However, substances produced during acid hydrolysis such as furfurals and HMFs act as intermediates in the production of AD inhibitors such as levulinate, formate, acetate, and various uronic acids, which prevent biogas production, when accumulate at high concentrations [46]. Alkaline hydrolysis may not have led to an increase in the concentration of free sugars, but enhanced the disruption of the lignocellulosic matrix, by lignin dissolution [45]. In the AD of the alkaline hydrolysate, the biogas release was delayed by several days mainly due to the lack of free sugars in the substrate, in contrast to the acidic hydrolysate, so that the microorganisms needed more time to adapt and degrade the carbon sources. Comparing the results of the current study with the international literature, it is found that the biogas yield of alkaline hydrolysate AD is lower, in contrast to the results of Ikeda et al. [44], in which AD of alkaline pretreated SMS (2% NaOH or KOH and heating at 80 °C for 30 min) resulted in higher biomethane yield. An even higher concentration of sodium hydroxide (10% NaOH at 40 °C for 24 h) was able to increase biogas yield, according to Sambusiti et al. [105].

Anaerobic co-digestion of cattle manure and SMS, in contrast to co-digestion of cattle manure and untreated wheat straw, increased biogas production, showing the importance of prior cultivation of fungi in wheat straw [71]. In their study, Gao et al. [76] observed that VS ratios SMS:DM = 1:1 or = 1:2 resulted in significantly higher biomethane production, while the literature confirms that increasing the ratio of feedstock to manure leads to failure of the digestion process in AD of lignocellulosic substances [72,74]. Among the different livestock manures (DM, Chicken Manure—CM, Pig Manure—PM) tested, the VS of SMS:CM = 1:2 ratio gave the most remarkable results (CBY ≈ 185 L/kg VS, MC ≈ 65%), in a 30-day AD [76]. Increasing the total solids content of AD and the VS of SMS to manure ratio equal to 1:9 led to an increase in CBY [58,74]; therefore, a corresponding ratio could increase biogas production in our case as well. Nevertheless, an increasing TS content led to a decrease in biogas yield [58,74]. AD under thermophilic conditions (≈55 °C) slightly increased the methane production during anaerobic co-digestion of SMS and deer manure, showing that AD at higher temperatures enhance biogas productivity [66,75].

## 4. Conclusions

Cultivation of edible mushrooms is an ever-emerging agro-food industry worldwide, with valuable widely accepted products, but also a significant volume of lignocellulosic bioresidues. These biowastes could be biotechnologically valorized in various ways. In the current study, the pretreatment of SMS and its successive anaerobic digestion were investigated. The co-digestion of the pretreated SMS with cattle manure enhanced the methane production for both acidic and alkaline hydrolysates. The chemical hydrolysis of SMS helps the assimilation of its components (mainly referred to holocellulose) during AD. The release of monosaccharides, due to acidic hydrolysis, combined with hydrothermal treatment, favors the rapid growth of microorganisms because of the immediate availability of a carbon source, in contrast to the low concentration of free sugars in alkaline hydrolysate and the subsequent delay in biogas production. Nevertheless, the disruption of the lignocellulosic matrix through alkaline treatment could yield a higher biogas production in a longer AD. Based on the results, SMS can be a substrate for anaerobic digestion, when pretreated with both chemical and alkaline hydrothermal hydrolysis.

## Figures and Tables

**Figure 1 microorganisms-11-00532-f001:**
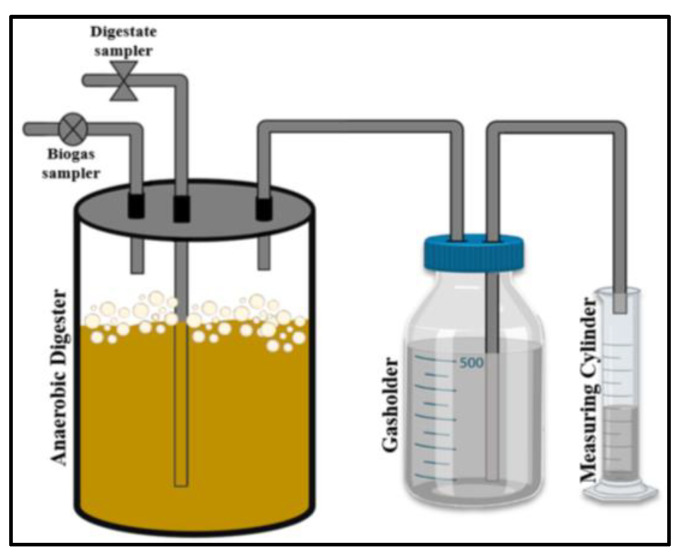
Representation of the bench-scale improvised digesters, which were used for anaerobic digestion of SMS hydrolysate.

**Figure 2 microorganisms-11-00532-f002:**
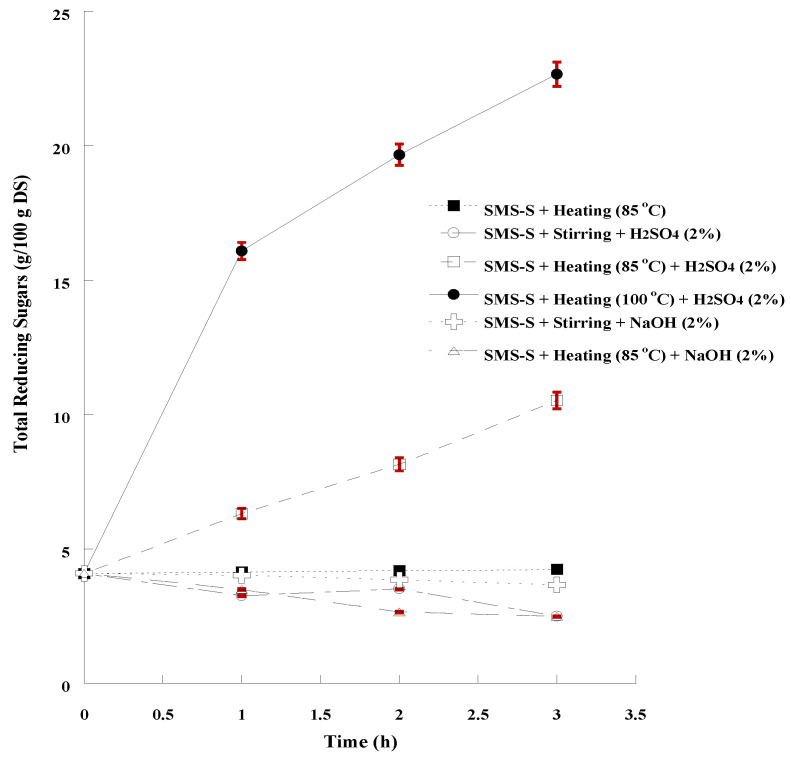
Monitoring the progress of SMS hydrolysis by determining the concentration of total reducing sugars in 3-h long processes. The presented results are the mean of two independent analyses.

**Figure 3 microorganisms-11-00532-f003:**
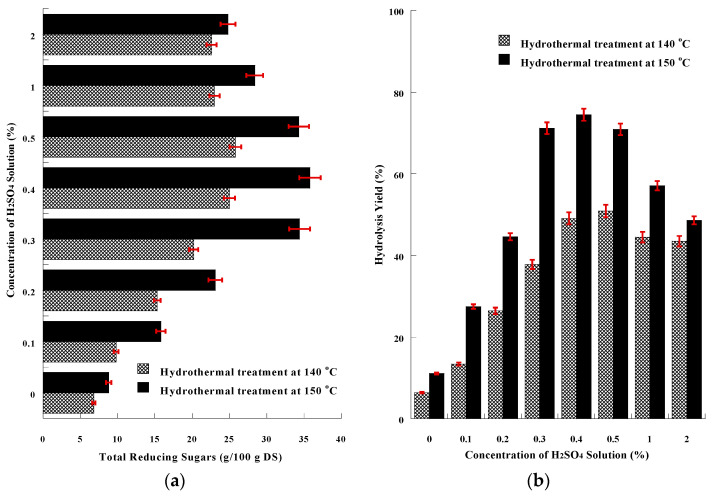
(**a**) Graphical presentation of Total Reducing Sugars (g/L) resulting from the hydrolytic combination of acidic chemical treatment (using various concentrations of acidic), and hydrothermal treatment at 140 or 150 °C. (**b**) Graphical presentation of hydrolysis yield (%) of acidic chemical treatment (using various concentrations of acidic), combined with hydrothermal treatment at 140 or 150 °C. The presented results are the mean of three independent analyses.

**Figure 4 microorganisms-11-00532-f004:**
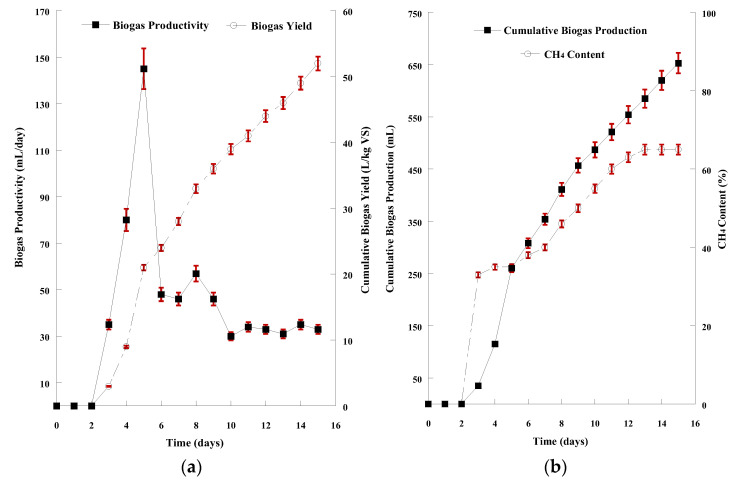
(**a**) Graphical representation of biogas productivity (mL/d) and biogas yield (L/kg VS) resulted from the anaerobic digestion of SMS acidic hydrolysate; (**b**) Graphical representation of cumulative biogas production (mL) and CH_4_ content of the biogas resulted from the anaerobic digestion of SMS acidic hydrolysate. The presented results are the mean of two independent digestions.

**Figure 5 microorganisms-11-00532-f005:**
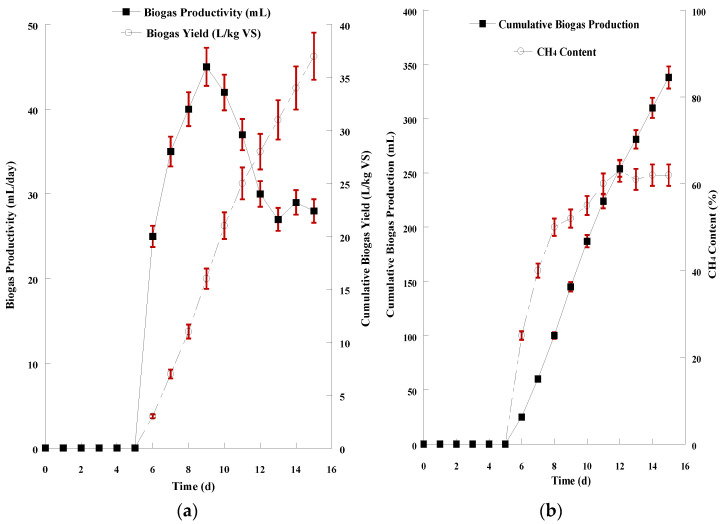
(**a**) Graphical representation of biogas productivity (mL/d) and biogas yield (L/kg VS) resulted from the anaerobic digestion of SMS alkaline hydrolysate; (**b**) Graphical representation of cumulative biogas production (mL) and CH_4_ content of the biogas resulted from the anaerobic digestion of SMS alkaline hydrolysate. The presented results are the mean of two independent digestions.

**Table 1 microorganisms-11-00532-t001:** Description of the applied pretreatment methods.

Treatment	Description
SMS-S + Heating (85 °C)	15 g SMS and dH_2_O in 200 mL total volume, plus heating at 85 °C in waterbath
SMS-S + Stirring + H_2_SO_4_ (2%, *v*/*v*)	15 g SMS, 4 mL H_2_SO_4_ and dH_2_O in 200 mL total volume, plus stirring
SMS-S + Heating (85 °C) + H_2_SO_4_ (2%, *v*/*v*)	15 g SMS, 4 mL H_2_SO_4_ and dH_2_O in 200 mL total volume, plus heating at 85 °C in waterbath
SMS-S + Heating (100 °C) + H_2_SO_4_ (2%, *v*/*v*)	15 g SMS, 4 mL H_2_SO_4_ and dH_2_O in 200 mL total volume, plus heating at 100 °C in waterbath
SMS-S + Stirring + NaOH (2%, *w*/*v*)	15 g SMS, 4 g NaOH and dH_2_O in 200 mL total volume, plus stirring
SMS-S + Heating (85 °C) + NaOH (2%, *w*/*v*)	15 g SMS, 4 g NaOH and dH_2_O in 200 mL total volume, plus heating at 85 °C in waterbath

**Table 2 microorganisms-11-00532-t002:** Description of the applied pretreatment methods during optimization of the hydrolytic processes.

Treatment	Description
SMS-S + HT (140 or 150 °C)	15 g SMS and dH_2_O in 200 mL total volume, plus hydrothermal treatment at 140 or 150 °C for 1 h in autoclave
SMS-S + HT (140 or 150 °C) + H_2_SO_4_ (0.1, 0.2, 0.3, 0.4, 0.5, 1 or 2%, *v*/*v*)	15 g SMS, 0.2, 0.4, 0.6, 0.8, 1.0, 2.0 or 4.0 mL H_2_SO_4_ and dH_2_O in 200 mL total volume, plus hydrothermal treatment at 140 or 150 °C for 1 h in autoclave
SMS-S + HT (140 or 150 °C) + NaOH (1.0 or 2.0%, *w*/*v*)	15 g SMS, 2.0 or 4.0 g NaOH and dH_2_O in 200 mL total volume, plus hydrothermal treatment at 140 or 150 °C for 1 h in autoclave

**Table 3 microorganisms-11-00532-t003:** Composition of dry SMS (spent mushroom substrate) of *P. ostreatus* cultivation. The presented results are the mean of two independent SMS samples.

Component	Content (g/100 g DS)
Free sugars	4.09 ± 0.21
Total protein	5.21 ± 0.34
Lipophilic substances	14.46 ± 0.62
Total lignin	13.77 ± 1.54
Cellulose	32.18 ± 2.81
Hemi-cellulose	10.45 ± 0.95
Ash	19.85 ± 0.86

**Table 4 microorganisms-11-00532-t004:** Quantitative data originated from various combinations of thermal and/or chemical hydrolysis of SMS. The presented results are the mean of two independent analyses.

Treatment	Total Reducing Sugars (g/100 g DS)	Released Reducing Sugars (g/100 g DS)	Hydrolysis Yield (%)
SMS-S + Heating (85 °C)	4.24 ± 0.12	0.15 ± 0.02	0.40 ± 0.02
SMS-S + Stirring + H_2_SO_4_ (2%, *v*/*v*)	2.49 ± 0.09	−1.60 ± 0.14	−3.80 ± 0.28
SMS-S + Heating (85 °C) + H_2_SO_4_ (2%, *v*/*v*)	10.53 ± 0.35	6.44 ± 0.55	15.10 ± 1.31
SMS-S + Heating (100 °C) + H_2_SO_4_ (2%, *v*/*v*)	22.67 ± 1.02	18.58 ± 1.10	43.60 ± 2.15
SMS-S + Stirring + NaOH (2%, *w*/*v*)	3.66 ± 0.09	−0.43 ± 0.02	−1.05 ± 0.07
SMS-S + Heating (85 °C) + NaOH (2%, *w*/*v*)	2.49 ± 0.31	−1.60 ± 0.08	−3.81 ± 0.27

## Data Availability

All data are presented in the text.

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
