# Peer review of "Spent Mushroom Substrate Hydrolysis and Utilization as Potential Alternative Feedstock for Anaerobic Co-Digestion"

_microorganisms, 2023, doi:10.3390/microorganisms11020532_

Round 1
Reviewer 1 Report
Nice study but a limited novelty. Reasonable amout of experimental work with the results are as expected from previously conducted similar studies on relevant substrates. See for example, the study of Dr. Shulin Chen in the first two refs below. Moreover. co-digestion is known to overcome issues related to inhibition, particularly from ammonia when digesting nitrogen-rich feedstock. The CH4 measurements are not by GC and CO2 absorption is not very accurate.
- Chen, S.; Liao, W.; Liu, C.; Wen, Z.; Kincaid, R.L.; Harrison, J.H.; Elliott, D.C.; Brown, M.D.; Solana, A.E.; Stevens, D.J. Value-Added Chemicals From Animal Manure. Northwest Bioproducts Research Institute, 2003. Available online: http://www.adktroutguide.com/files/Elliott_PNNL_value_from_manure.pdf
- Liao, W.; Liu, Y.; Liu, C.; Chen, S. Optimizing dilute acid hydrolysis of hemicellulose in a nitrogen-rich cellulosic material—Dairy manure. Bioresour. Technol. 2004, 94, 33–41.
- Wen, Z.; Liao, W.; Chen, S. Hydrolysis of animal manure lignocellulosics for reducing sugar production. Bioresour. Technol. 2004, 91, 31–39.
- Yang, Q.; Wang, H.; Larson, R.; Runge, T. Comparative study of chemical pretreatments of dairy manure for enhanced biomethane production. BioResources 2017, 12, 7363–7375. Available online: https://ojs.cnr.ncsu.edu/index.php/BioRes/article/view/BioRes_12_4_7363_Yang_Chemical_Pretreatments_Dairy_Manure/5522
- un, Y.; Cheng, J. Hydrolysis of lignocellulosic materials for ethanol production: A review. Bioresour. Technol. 2002, 83, 1–11.
- Taherzadeh, M.J.; Karimi, K. Pretreatment of lignocellulosic wastes to improve ethanol and biogas production: A review. Int. J. Mol. Sci. 2008, 9, 1621–1651.
Abstract
The abstract should be rewritten. line 12-18 could be shortened to one or two lines. Focus on what you did and present your results briefly. Simply, say this study evaluated (state the number) alternative hydrolysis techniques and biogas yield from the best one. Then move on to present the numerical results.
You use the comma "," excessively to the extent that it ruined your presentation. L12-4 (4 commas), L14-17 (4 comas), L18-21 (3 commas), L21-26 (7 commas), L26-28 (3 commas).
Introduction
L 60-64 (10 commas)! L65-69 (11 commas)! I stopped counting here. Please fix the issue of commas.
L39 change "by" to "from"
L41, I don't know why this statement is here. Fungi-based biological treatment of what and for what purpose? Do you mean for hydrolyzing lignocellulosic material? if yes, then this is far from your work.
L47-51 Please rewrite and check the English. I am not comfortable with "against inflammation" and the rest of the sentence.
L51-52 why the capitalization of "Industrial, Green, and Red Biotechnology"?
L96-98 check the English language. What comes after but has to contradict what comes before it. The statement does not make sense. Biogas has been studied extensively for both mono- and co-digestion of substrates.
L104 "was used as feedstock" for what? also, add "a" before "feedstock"
L106 please delete "critically"
Material and Methods
rewrite "to weight constancy" to "until constant weight"
L190-193 and L209-214: these experimental designs need to be shown as Tables. Make the tables stand alone and give a sufficient explanation of the samples and conditions.
Figure 1: why is this figure so large while it does not provide much or necessary info?
Results - Discussion
L325 subscripts in H2SO4
L398 change "&" to "and"
in many places in the manuscript, the unit of a kilogram (kg" is written with a capital (upper case) "K". Units are written with small letters (lower case), and thus kilogram unit is "kg".
Reviewer 2 Report
The main question addressed by this specific research is relevant and interesting. Increasing demand for energy is a major issue in our society. Therefore, the search of alternatives for sustainable forms of bioenergy production is necessary due to the depletion of fossil fuels and the worsening climate change caused by greenhouse gas emissions. In this manuscript authors describe the optimization of spent mushroom substrate hydrolysis (SMS) and anaerobic digestion of the hydrolysates for biomethane production.
Authors have found an effective method for pre-treatment of SMS and optimised the anaerobic process, where cattle manure was used for co-digestion. The disruption of lignocellulosic matrix after alkaline treatment seems to be a promising method for reaching higher biogas production, therefore author’s contribution could be interested for the target readership. It shows potential use in agro-food industry.
Paper is well written, text is clear and easy to read. Nevertheless, I still have some comments and suggestions.
Title: properly reflects the subject of the paper.
Abstract: provides an accessible summary of the paper.
Keywords: they accurately reflect the content. My comment is that normally we do not use abbreviations as the keywords, thus, authors are asked to replace SMS with the complete words.
Introduction: Authors described the problem, they summarized recent research related to the topic, clearly demonstrated the need for investigations in the area of efficient biogas production and explained why this research was carried out.
Some more comments:
Line 53: FAOSTAT – please, explaine this abbreviation
Line 58 and further: Use “kg” and not “Kg”
Authors should write units according to SI standards and symbols. E.g.: hour-h, day-d.....
Line 104: Authors should check this sentence (The SMS hydrolysate.....) because it is not clearly written.
Materials and methods
In this section authors gave enough data that I can conclude that the described research work is repeatable. The same experiments could be repeated by other researchers. But I have the question regarding Statistical analyses. I haven’t found anything about this. Were the experiments done in duplicates? What was standard deviation?
Equations for Ash, Volatile solids, Sugar concentrations..... should not be inserted into the text. Authors are asked to place them separately.
Results and discussion:
Obtained results are clearly discussed.
Line 347: Authors wrote: “The kinetics of the acidic hydrolysate for 15 days AD is presented in Figures 4a and 4b”.
Kinetics concerns with understanding the rates of chemical reactions. Chemical kinetics includes investigations of how experimental conditions influence the speed of a reaction and gives information about the reaction’s mechanisms, as well as the construction of mathematical models that also can describe the characteristics of a chemical reaction. Therefore, constructing the graph: Biogas productivity versus AD time can not be classified as kinetics. Authors are asked to change the corresponding text in the Result section or additionally to develop a kinetic model of anaerobic digestion.
Figures in general are clear, but I suggest bigger letters for labelling both axes and reducing the sizes of curves labels.
2Figures 3, 4 and 5 should be reduced if they will stay in present form or graphs a) and b) should be presented as separate figures.
TTables 1 and 2 are clear.
Conclusions: are consistent with the evidence and arguments presented in the manuscript and they address the main question posed. The main result of this research work is that SMS can be used as a substrate for anaerobic digestion when pre-treated with both chemical and alkaline hydrothermal hydrolysis. In that case biogas production could be even increased.
Reviewer 3 Report
Dear Editor: This paper can be published your journal.
Author Response
Thank you very much.
Round 2
Reviewer 2 Report
Thank you for considering all my comments.